# Vitis OneGenE: A Causality-Based Approach to Generate Gene Networks in *Vitis vinifera* Sheds Light on the Laccase and Dirigent Gene Families

**DOI:** 10.3390/biom11121744

**Published:** 2021-11-23

**Authors:** Stefania Pilati, Giulia Malacarne, David Navarro-Payá, Gabriele Tomè, Laura Riscica, Valter Cavecchia, José Tomás Matus, Claudio Moser, Enrico Blanzieri

**Affiliations:** 1Research and Innovation Centre, Department of Genomics and Biology of Fruit Crops, Fondazione Edmund Mach, 38098 San Michele all’Adige, Italy; giulia.malacarne@fmach.it (G.M.); claudio.moser@fmach.it (C.M.); 2Institute for Integrative Systems Biology (I2SysBio), Universitat de València-CSIC, 46908 Paterna, Valencia, Spain; david.navarro.paya@gmail.com (D.N.-P.); tomas.matus@uv.es (J.T.M.); 3Centre for Integrative Biology (CIBIO), University of Trento, 38123 Trento, Italy; gabriele.tome@studenti.unitn.it; 4Department of Information Engineering and Computer Science, University of Trento, 38123 Trento, Italy; laura.riscica@studenti.unitn.it (L.R.); enrico.blanzieri@unitn.it (E.B.); 5CNR-Institute of Materials for Electronics and Magnetism, 38123 Trento, Italy; valter.cavecchia@imem.cnr.it

**Keywords:** OneGenE, stilbenoids, lignin, causal inference, gene network, transcriptomics, peroxidase, laccase, dirigent protein

## Abstract

The abundance of transcriptomic data and the development of causal inference methods have paved the way for gene network analyses in grapevine. Vitis OneGenE is a transcriptomic data mining tool that finds direct correlations between genes, thus producing association networks. As a proof of concept, the stilbene synthase gene regulatory network obtained with OneGenE has been compared with published co-expression analysis and experimental data, including cistrome data for MYB stilbenoid regulators. As a case study, the two secondary metabolism pathways of stilbenoids and lignin synthesis were explored. Several isoforms of laccase, peroxidase, and dirigent protein genes, putatively involved in the final oxidative oligomerization steps, were identified as specifically belonging to either one of these pathways. Manual curation of the predicted sequences exploiting the last available genome assembly, and the integration of phylogenetic and OneGenE analyses, identified a group of laccases exclusively present in grapevine and related to stilbenoids. Here we show how network analysis by OneGenE can accelerate knowledge discovery by suggesting new candidates for functional characterization and application in breeding programs.

## 1. Introduction

The availability of an ever-increasing number of *Vitis* ssp. transcriptomic data and the creation and maintenance of integrated and comprehensive compendia [1,2] are paving the way to developing system biology approaches in the grapevine. Reusing omics data, in line with the GO FAIR guiding principles [3], represents an advantageous opportunity to drive knowledge discovery from existing data. In particular, gene network reconstruction is receiving growing attention. Gene regulatory networks (GRNs) provide a convenient way of representing as graphs the functional interactions (as edges) of the genes (as nodes) of an organism [4]. Besides highlighting gene correlations, this representation can contribute to hub gene identification, gene function prediction, data integration from different omics techniques, and comparative gene network analyses across species [5]. One approach to finding correlations among genes is based on the “guilt by association” assumption that genes with similar expression profiles are likely to be functionally related. This approach, primarily based on Pearson’s correlation and mutual information analyses, produces weighted gene co-expression networks (WGCNs). In grapevine, WGCN analysis has been applied in many cases, to analyze a single or a few experiments [6,7,8,9,10,11,12]. Recently, an improved version based on the aggregation of multiple WGCN analyses performed on different datasets (AggWGCN) has been proposed to identify gene networks using a selection of public transcriptomic data [13,14]. For co-expression analyses, the sample size and quality—the number and the kind of experiments selected to build the dataset—are important features to consider for the algorithm performance.

A different approach is represented by causal discovery analyzing observational data or data collected on various conditions. The search through causal relationships among co-related variables (here transcript expression levels) aims to go beyond simple correlation, simplifying the correlation graph by deleting relationships that appear to be independent once a separation set of variables is applied. A notable example of this approach is the PC algorithm [15], whose application to transcriptomic data could be done by iteratively subsetting the variables and combining the results afterward. This approach is adopted by NES^2^RA; a causal inference method applied to the entire VESPUCCI database to expand known local gene networks (LGNs) [16]. The high number of PC-algorithm runs required by the iterative subsetting is computed in a distributed way by the gene@home project that exploits the BOINC platform for volunteer computation [17]. NES^2^RA produces an association network whose edges represent putative direct interactions between genes, as an interaction still exists despite a high number of attempts to remove it by individuating a separation set of variables.

Despite NES^2^RA’s reported ability to identify gene interactions starting from a nondedicated large transcriptomic dataset, some limitations related to its practical use were, however, evident. First, the heavy computation of the LGN expansion cannot be requested directly by the user and may require up to several days to complete, even with the use of the BOINC distributed computing platform. Second, the analyses are not reusable, as they are very likely to be unique elaborations, given that the number of possible gene combinations is exponential in the size of a genome. Finally, a priori knowledge of the LGN is not always the starting point of an inquiry; researchers often have good candidate genes and need to first refine the LGN before expanding it. To overcome these difficulties, in the present work, we adopted the OneGenE approach, which aims to expand each single gene in an organism. OneGenE systematically runs-single gene NES^2^RA expansions with fixed parameters and combines them afterward to simulate LGN expansions. In this way, the single-gene expansions can be used multiple times, thus effectively reducing the computational effort [18].

Stilbenoids are a group of polyphenols acting as phytoalexins in grapevine and other species due to their antifungal properties [19]. They derive from the STS-catalyzed synthesis of trans-resveratrol, which can be further modified by enzyme-mediated methylation, hydroxylation, glycosylation, or oligomerization [20]. Stibenoid synthesis often occurs upon biotic stresses, and oligomers show higher toxicity than monomers [20,21]. The STS of the grapevine gene family has been characterized as encompassing 41 predicted isoforms falling into three main phylogenetically distinct groups, specifically group A clustered on chromosome 10 and groups B and C on chromosome 16 [22]. The high copy number of STS genes in the grapevine genome suggests a high dosage requirement of the encoded protein as well as a functional specialization of the different isoforms. It is known that resveratrol accumulation in grapevine is strictly controlled at the transcriptional level by the regulation of STS isoforms expression, specifically activated during development [23] and in response to biotic and abiotic stresses [22,24,25]. The first study aiming to discover the *VviSTS* regulators has identified two R2R3-MYB factors, *VviMYB14* and *VviMYB15*, explicitly activating the promoters of *VviSTS41* and *VviSTS29* genes in transient reporter assays [25]. Other regulators have been identified since then, and we refer the reader to the Results section, where they are presented in detail to support the validation of the approach. Neither the enzymes catalyzing resveratrol oxidative polymerization into dimers or higher degree oligomers, nor their transcriptional regulators, have so far been identified and characterized. Interestingly, six peroxidase (PRX) genes (*VviPrxIII08a*, *VviPrxIII08b*, *VviPrxIII15a*, *VviPrxIII21a*, *VviPrxIII23a*, and *VviPrxIII34a*) and two laccase (*LAC)* genes (VIT_218s0001g02400 and VIT_218s0001g02410) were proposed as possibly involved, based on their localization in QTL regions related to monomeric and oligomeric stilbenoid and ε-viniferin accumulation, respectively [26]. Moreover, the involvement of dirigent proteins (DIR) has been suggested due to the stereospecificity of stilbene oligomers, but it has not been proven so far [21,27].

Lignin is a phenylpropanoid-derived polymer found in specific cell types of vascular plants, particularly those with secondary thickened cell walls involved in the transport of water or mechanical support. Lignin is derived from the oxidative bimolecular radical coupling of three monolignols— coniferyl, sinapyl, and p-coumaryl alcohols—in a reaction catalyzed by LAC and class III PRX [28] and assisted by DIR proteins, which ensure the stereoselectivity of the reaction [29,30]. The three monolignols are synthesized through a branch of the phenylpropanoid pathway, initiated by the enzymes cinnamoyl CoA reductase (CCR) and hydroxycinnamoyl transferase (HCT). Different members of the PRX III gene family have been functionally characterized as associated with lignification in *Arabidopsis* [28,31]. Phylogenetic reconstruction of the grapevine *PRX* gene family, together with the one from *Arabidopsis*, allowed us to identify the grapevine putative orthologs of *AtPrx* members involved in lignification [26]. DIR families are widespread in most plant species, from herbaceous to perennials, monocots, and dicots [29,30]. However, it has not been characterized yet in the grapevine. The biosynthesis of lignin is transcriptionally controlled through the regulation of genes encoding enzymes involved in monolignol biosynthesis and lignin polymerization. High-level regulators are members of the NAC and MYB families, as elucidated in *Arabidopsis*; however, identification of the whole regulatory cascade is still ongoing [32].

In the grapevine, some putative R2R2 MYB TFs involved in lignin biosynthesis have been identified by GCN and phylogenetic analyses based on *Arabidopsis* comparison [33]. The *VviMYB103* (VIT_02s0025g00320) and *VviMYB26 (*VIT_09s0002g01670) were identified as the putative orthologs of *AtMYB103*, positively regulating ferulate 5-hydroxylase (F5H) expression and syringyl lignin biosynthesis in the stem [34], and *AtMYB26* involved in the formation of secondary thickening in the anther endothecium [35]. *VviMYB190* (VIT_00s1241g00010) and *VviMYB191 (*VIT_16s0022g01210) clustered with *AtMYB85*, associated with the regulation of lignin biosynthesis [36]. Finally, *VviMYB185* (VIT_06s0004g02110) and *VviMYB186* (VIT_13s0064g00960) belong to the same subgroup of *AtMYB46*, associated with the regulation of cell wall biosynthesis [37]. The six MYBs putatively related to lignin synthesis have been used as input genes for OneGenE to identify LAC, PRX, and DIR isoforms possibly involved in lignin synthesis.

In the present work, first, we offer a proof of concept for the OneGeneE approach by reconstructing the well-characterized GRN of grapevine STS and comparing our results with the state-of-the-art available knowledge. We also present the OneGenE website, where the expansion lists for any grapevine gene are made available and describe the procedure required to generate association networks starting from a set of genes. Second, we present the application of the OneGenE method to two uncharacterized grapevine pathways—the stilbenoids and lignin biosynthesis—to show the potential of the OneGeneE approach. Finally, the results obtained with OneGenE are integrated with the phylogenetic analysis of the *Vitis* LAC and DIR families shedding light on group functional specialization within these families.

## 2. Materials and Methods

### 2.1. Computation of OneGenE Expansion Lists

The expansion of each gene of the *Vitis vinifera* genome originated from the run of the algorithms of the OneGenE system [18,38]. Each expansion procedure consisted of 2000 iterations of our C++ implementation (https://bitbucket.org/francesco-asnicar/pc-boinc/, last accessed on 16 June 2020) of the PC-algorithm skeleton procedure [39] (α = 0.05) to 29 sets of 1000 variables (genes), which included the gene to be expanded, and a random subset of 999 genes sampled without replacement. The blocks scheme of the OneGenE architecture is shown in Blanzieri et al. 2020, Figure 1 [18]. The input data were extracted from the 28,013 × 1131 normalized expression data matrix initially obtained from the VESPUCCI repository [1], filtered, and preprocessed [16]. Each expansion list is ordered with respect to the relative frequency, namely F_rel = #times the gene is present in the output of the PC-algorithm/# times the gene is present in the input of the PC-algorithm. Overall, the computation of the expansion lists of all the genes required 28,013 × 29 × 2000 = 1,624,754,000 runs of the PC algorithm, each run taking 5.63 s on average on our reference machine (Intel i7-4770K, Ubuntu). Therefore, the computation was done within the gene@home project on the volunteer distributed computation platform TN-Grid (20 TeraFLOPS on average) [17], powered by BOINC [40] software.

The OneGenE expansion lists were annotated using the catalogue.INTEGRAPEv2 (http://www.integrape.eu/index.php/resources/genes-genomes, 19 October 2021), published works of genes families already characterized in grapevine, Vitisnet gene annotation [41], and the VCost.v3 annotation [42]. The “gene clusters”, as defined in the VESPUCCI database [1], could not be automatically annotated.

### 2.2. Programs Developed to Reconstruct Gene Networks Using OneGenE Output

Three programs have been developed in Python v3.8.10 (https://www.python.org/) to create, expand, visualize, and analyze gene networks starting from the OneGenE expansion lists. The first two tools have been implemented on the website (http://vitis.onegenexp.eu).

#### 2.2.1. Create Network

This program computes the interactions within a set of input genes by analyzing their OneGenE expansion lists. As input, it takes a compressed file (.zip) containing the OneGenE expansion lists, downloaded from the OneGenE website (vitis.onegenexp.eu). The user must set a minimum threshold of relative frequency, which defines the number of genes considered for the analysis. The analysis is performed in pairs: for each pair of input genes, the mutual presence of each gene in the trimmed expansion list of the other one is verified, and the mean of their relative frequency values is computed. The tool produces two tables, one for the edges/interactions and one for the nodes/genes, an image file with a static representation of the gene network as an undirected weighted graph, and a .json file which can be imported into Cytoscape 3.8.2 [43] or Cytoscape.js [44] for improved visualization of the gene network.

#### 2.2.2. Expand Network

This program aggregates the OneGenE expansion lists of a set of input genes according to the criteria available to the user. In addition to the compressed file with the expansion lists, it requires a text file input with the V1 IDs of the genes to be aggregated. The aggregation can be performed according to the following four criteria: (i) by considering the genes above a threshold of relative frequency (-frel -f) or (ii) of rank value (-rank); (iii) by considering the genes belonging to a functional category according to the Vitisnet annotation (-pattern -f), for example, setting “-pattern vv60” will aggregate the genes annotated as transcription factors present in the expansion lists; (iv) by considering the genes shared among the input genes (-shared -f). The output will be composed of the same types of files explained for the first tool (“Create network”).

#### 2.2.3. Network Analysis

This program performs Gene Ontology (GO) and promoter motifs enrichment analyses. By choosing the “-topGO” option, the program performs an enrichment analysis based on the Biological Process and Molecular Function categories of the GO, using the TopGO Bioconductor package [45]. It requires a text file with the V1 ids of the genes to be analyzed as input and a GO annotation file of the whole *V. vinifera* genome. The output consists of a table and a graphical representation of the enriched categories, ranked according to their statistical significance. By choosing the “-DREME” option, the program will prepare a file containing the 1-kb promoters of the genes in FASTA format. This file is suitable for promoter motifs enrichment analysis using the STREME tool of MEME suite [46]. The program requires the same text files input with the V1 IDs of the genes to be analyzed and a file containing the 1-kb promoters of the whole grapevine genome (12X.v1).

### 2.3. Genome-Wide Search and Gene Model Curation of LAC and DIR Gene Families

To retrieve the complete list of isoforms for the *LAC* and *DIR* gene families, HMMER search based on Pfam domains was performed on the 12X.v2 genome assembly (VCost.v3 annotation) [42]. Three different multicopper oxidase Pfam domains (PF00394, PF007731, PF07732) and the dirigent Pfam domain (PF03018) were used for *LAC* and *DIR* identification, respectively. The genes containing at least one Pfam domain with an e-value ≤ 10^−5^ were treated as potential *LAC* or *DIR* isoforms. The Web Apollo genome editing platform [47] was used to manually curate their gene models based on public RNA-seq data, as a correct coding sequence was a prerequisite for a correct phylogenetic analysis.

### 2.4. Phylogenetic Analysis of the LAC and DIR Gene Families

The protein sequences of the manually curated genes were then subjected to phylogenetic analysis, together with *Arabidopsis* genes. For the LAC analysis, the 17 LACs present in the *Arabidopsis* genome were included as well as the selected members of other multicopper oxidase families sharing similar copper-centered protein domains, such as Ascorbate Oxidases (AOs) and SKU5 Similar (SKS) proteins. Nitrite reductases (NIRs) are multicopper oxidases, but an initial inspection involving multiple protein alignments showed that none of the potential *V. vinifera LAC* genes appeared to correspond to this multicopper oxidase family. Hence, members of this family from *Arabidopsis* were not included in the final phylogenetic analysis. Low Phosphate Root (LPR) multicopper oxidases were used as an outgroup. The complete list of *Arabidopsis* and *Vitis* genes used for the phylogenetic analysis is available in Appendix A. The molecular evolution model selected for LAC analysis was ‘WAG + F + G’ for the bayesian phylogenetic inference carried out with MrBayes [48]. A total of three million generations were run, reaching an average standard deviation of split frequencies of 0.010 and full chain convergence. For the DIR family phylogenetic analysis, all 25 *Arabidopsis* DIR proteins were included (Appendix A), and the model selected was ‘JTT + G’. A total of one million generations were run, reaching an average standard deviation of split frequencies of 0.0075 and full chain convergence.

### 2.5. Data and Code Availability

OneGenE annotated lists can be downloaded as compressed files from the website vitis.onegenexp.eu. The reconstructed networks computed on the website can be downloaded as well, both as images and in a Cytoscape-compliant format. The programs developed in python to reconstruct and analyze the gene networks are available at https://github.com/gabrieletome/vitis_tools (last updated on 1 November 2021). In addition to the code, the repository contains the files used for the analyses presented in the present article as examples.

## 3. Results

### 3.1. The Grapevine OneGenE Website

In order to make OneGenE results available to the community, we built a website, vitis.onegenexp.eu (last update on 19 November 2021). From the “OneGeneE analysis” page, the user can perform a query with a list of genes of interest, using the gene IDs of the 12X.v1 release of the grapevine genome annotation, and download the corresponding expansion lists from the result page. The “Network analysis” link redirects the user to a new page where they have options to either build the network formed by the selected genes (“Create network”) or to expand the obtained list to find new interacting genes (“Expand network”). The networks are visualized in the bottom part of the page, with different layouts and the choice of using different IDs for the nodes (12X.v1, VCost.v3, gene symbol). These networks can be exported as images (.png) or as files (.json) for further customization, for example, in Cytoscape [43]. A “Tutorial” page has been created with the STS network analysis described in the present paper as an example.

### 3.2. The Stilbene Synthase Gene Regulatory Network as a Proof of Concept

The reconstruction of the STS GRN represented a suitable case for validating the OneGenE approach because of its prevalent transcriptional regulation and the availability of comparative published studies. The strategy we applied is summarized in Figure 1 and represents a general model also used for the stilbenoid and lignin networks. The process starts with a query on the OneGenE website to retrieve the expansion lists of the input genes, using them to build their network. This network is then expanded by consolidating all the lists into one aggregated list. Although, in principle, this new network can be visualized, too many nodes make the representation less clear. Alternatively, the user can inspect this list and choose the newly identified genes of interest, such as specific transcription factors (TFs) or gene families. This restricted set of genes can be used to query the website and build a network, as previously described. Gene networks are evaluated through enrichment analysis of GO functional categories to provide a general idea of the gene-sets obtained with the expansion analysis.

A query with the 41 *VviSTS* 12X.v1 IDs allowed the retrieval of 22 expansion lists from the OneGenE website (Appendix A), as two isoforms were not present in the VESPUCCI processed dataset, and 21 isoforms were compressed into 4 clusters to overcome the problem of probe mapping ambiguity [1]. These lists were used to create the network formed by *VviSTS*s, setting a minimum relative frequency threshold of 0.5 to focus on the most direct connections (Appendix A). An almost fully connected network was obtained. Remarkably, the three *VviSTS* groups identified as A, B, and C in the phylogenetic analysis [22] did not show any mirroring modularity at this level of analysis. As a second step, the STS network was expanded to find its related genes (relative frequency threshold set at 0.5 as above). An aggregated list of 955 genes has been obtained (Appendix A), enriched in defense response, biosynthetic process, lignin catabolic process, and oxidation-reduction process, according to the GO biological process categories (Appendix A). To produce the STS GRN, the list was manually inspected to identify the TFs. A total of 72 TFs were found, 39 of which connected to at least two *VviSTS*. This STS GRN has been compared with available published data to validate OneGenE analysis results (Table 1).

*VviMYB14* and *VviMYB15*, initially identified by Höll et al. [25], were found as expected among the most connected to *VviSTS*. *VviWRKY24* and *29*, previously identified as co-expressed with *VviSTS* during drought stress [53], were also found by OneGenE, as were the 20 TFs previously identified by GCN analysis and promoter activation assay [49], belonging to the R2R3MYB, WRKY, NAC, bHLH, and ERF families. Recent functional characterization works in a wild grape confirmed the interaction of *VqWRKY53* with *VqMYB14* and *VqMYB15,* affecting *VqSTSs* activation in *V. quinquangularis* [51]. A recent transcriptomic analysis—not included in VESPUCCI and thus not used by OneGenE— proving the role of stilbenoid accumulation in plant defense to *Plasmopara viticola* [50], extensively validated our GRN, as well. Finally, a significant overlap with *VviMYB14* and *VviMYB15* cistrome analysis was observed [52].

Beyond providing suitable proof of concept, the STS GRN here obtained presented several interesting features worth mentioning. The visual representation of the network highlighted a partitioning of *VviSTSs* around three hubs, namely *VviMYB14*, the *VviWRKY43*-*53* pair, and the *VviMYB13-15-VviWRKY24* triad (Figure 2A). Again, the three groups of *VviSTSs* appear largely overlapping, with *VviMYB14* showing the highest degree of connection to almost all group B STSs. The present network is consistent with the reported activation of *VviSTS29* (here in CL_B_2) by *VviMYB14* and *VviMYB15*, alone and in combination with *VviWRKY3*, and by *VviWRKY24*, which have been experimentally demonstrated [25,49]. Moreover, the OneGenE approach identified four members of the Jasmonate ZIM domain (JAZ) TF family, all connected to *VviMYB14* (Figure 2B). This is the first time this family of regulators has been reported as associated with the STS GRN. Two genes encoding MYC-type TFs, *VvibHLH001-VviBIM3* and *VvibHLH066-VviMYC7E* (J.T. Matus, personal communication), have been identified too, as inversely related to specific *VviSTS* isoforms. The former is directly linked to *VviJAZ3*, while the latter is located in the *VviWRKY43*-*53* subnetwork. Further investigation is required to support their involvement in jasmonate signaling. Finally, two TFs belonging to the Zinc finger C3HC4 type, *ZFC3HC4_163* and *ZFC3HC4_36*, appeared highly connected within the *VviWRKY43-53* subnetwork, one of which was also identified by DAP-seq analysis [52].

### 3.3. Laccase, Peroxidase and Dirigent Proteins Isoforms Involved in the Stilbenoid and Lignin Networks

As a case study, the OneGenE method was applied to gain more insight into the stilbenoid and lignin pathways. We focused on the gene families putatively involved in the oligomerization of resveratrol and monolignols. Concerning the former, we used the same aggregated list derived from the STS network expansion: it included many genes annotated as LAC, PRX III, and DIR, as well as a glycosyl- and O-methyl-transferases, putatively involved in resveratrol conversion to piceid and pterostilbene (Appendix A, Figure 3). For the lignin network reconstruction, we used the six MYB TFs identified in [33] as a query on the OneGenE website to expand them, obtaining an aggregated list of 676 genes (Appendix A). They were enriched in the lignin catabolic process and regulation of nitrogen (Appendix A), supporting their involvement in lignin metabolism. Several genes annotated as LAC, PRX III, and DIR were found, as expected, in the network (Figure 4).

The first very evident outcome was that both networks included several isoforms of PRX III, LAC, and DIR but altogether distinguished. For PRX class III, phylogenetic analysis and a precise name assignment were already available [26]. This information allowed us to identify six PRX III of class 01—located on chromosome 1— in the stilbenoid network together with two PRXIII of class 22 and one of class 23. *VviSTS20* was connected to all three PRX III classes, the O-methyltransferase and the glycosyltransferase, the latter in a negative way, suggesting that this STS could be involved in pterostilbene accumulation and its oxidative modifications. *VviSTS20* also showed a particular regulation, as it was connected only to *VviJAZ4* and *ZFC3HC4_163* (Figure 2). *VviSTS38*, *32*, *47*, and *9* interacted with many class 01 PRX III and DIR and belonged to the *VviWRKY43-53* subnetwork. Finally, PRX III and DIR were located distantly in the network, interacting not directly but through STSs nodes, despite being assumed to catalyze consecutive steps of stilbenoids polymerization. Conversely, the lignin network looked more uniformly connected. OneGenE identified two PRX III: one of class 12 and one of class 07. The latter, homolog of AtPRX64, was highly connected to three of the MYB TFs, five LAC, and one DIR gene.

A thorough phylogenetic characterization of these families has been undertaken to better describe these networks and provide precise indications on the LAC and DIR isoforms putatively involved in one or the other pathways. A genome-wide search for LAC genes in the 12X.v2 genome assembly identified another 44 multicopper oxidases in addition to the 77 already annotated as LAC in the 12X.v1 (Table 2 and Appendix A). Manual curation of the 121 genes was required to correct the gene models and improve the phylogenetic analysis. Overall, 25 coding sequences (CDSs) were corrected through manual curation. After this step, another 25 genes were renamed multicopper oxidase-like genes (MCO-Like) due to their unusual length (less than 400 residues, while all the *Arabidopsis* genes ranged from 526 to 588 aa) and very low RNA-seq support. This analysis produced a final list of 95 multicopper oxidases, which were used for the phylogenetic analysis with the *Arabidopsis* genes (Appendix A and Figure 5).

The phylogenetic tree in Figure 5 clearly showed that 19 *Vitis* genes corresponded to multicopper oxidases such as low phosphate root (LPR) genes, ascorbate oxidases (AO), and SKU5 similar (SKS) proteins, while 76 genes were actual LAC isoforms. They could be divided into nine subgroups, 7 of which contained both *Vitis* and *Arabidopsis* genes and reproduced the groups already described for *Arabidopsis* [54,55], while groups 8 and 9 only contained *Vitis* LACs. Interestingly, the remarkable gene expansion of the grapevine LAC family was primarily due to the two additional *Vitis*-specific subgroups located on chromosome 18 (Table 3). Due to the lack of a clear correspondence between *Vitis* and *Arabidopsis* genes and these additional large groups, LAC isoforms were named according to gene position within the phylogenetic tree, as recommended in [56]. By doing so, the consecutive numbering of LACs from each subgroup was granted.

Even more interesting finding was that all the LACs identified as part of the stilbenoid pathway belonged to group 8, suggesting a gain of functional specificity of this group related to resveratrol oligomerization. Conversely, LACs related to the lignin network belonged to subgroups 1–3 and one to group 8. Within the LAC family, the functional subdivision almost entirely corresponded to a phylogenetic difference.

The same pipeline of analysis was applied to characterize the DIR family. The HMMER search on the 12X.v2 genome assembly identified another nine genes in addition to the 32 already annotated as DIR in the 12X.v1 (Table 4). Manual curation of the gene models improved the CDS prediction of eight isoforms.

The phylogenetic analysis performed on the 41 and 25 DIR isoforms from *Vitis* and *Arabidopsis*, respectively, showed that all putative DIR proteins from *Vitis* were closely related to an *Arabidopsis* homolog and could be considered true DIR proteins (Figure 6). The previously described DIR groups in *Arabidopsis thaliana* appear to be recapitulated in general [57,58], however, further subdivisions were suggested by the phylogenetic branches, and a numeric suffix was appended to the groups a, bd, and e accordingly. The bd group was clearly expanded in *Vitis* with respect to *Arabidopsis*. The final DIR nomenclature was defined according to gene position within the phylogenetic tree as done for LAC. The integration of the OneGenE functional results and the phylogenetic analysis produced less distinguished group associations, as the DIR isoform related to the stilbenoid network belonged to subgroup bd2 and the orphan *VviDIR41*. In contrast, those related to the lignin network belonged to subgroup a, bd1, and bd2. *VviDIR 12*, *13*, and *16* and were connected to almost all *VviSTSs*, while *VviDIR41* appeared to belong to a smaller group formed by *VviSTS45*, *7*, and the four clusters, which account for 21 isoforms.

## 4. Discussion

The power of applying the PC-based causal inference approach to grapevine transcriptomic data has already been appreciated when expanding local gene networks with NES^2^RA [16]. The strategy to start from a completely connected network and remove edges in two steps, the first based on Pearson’s correlation and the second based on conditional independence, allows not only to simplify a co-expression network but also to improve it by removing indirectly linked genes thus discriminating between co-occurrence and functional dependency. The main drawback of NES^2^RA for the biologist users was the impossibility to perform their expansion analyses themselves due to the time and computational effort requirement for each analysis and despite the availability of the BOINC-based distributed computation platform of TN-GRID [17]. Another drawback was the non-reusability of such results, being GRN specific. OneGenE overcomes these problems by expanding a local gene network made of one gene, computing it for all the genes of the genome, and making them readily available to the user, free to expand the preferred gene network by using the OneGenE lists (Figure 1). The OneGenE approach better accomplishes GO FAIR requirements [3], as the expansion precomputed lists are reusable and have been made Findable and Accessible through the website presented here. An example of Interoperability of the data is provided by the tools that we developed to reconstruct gene networks and have been implemented on the website analysis page to facilitate the usage of OneGenE lists. The architecture of the project is suitable for future improvement and implementation, such as on one side, the re-calculation of all the expansion lists on the upcoming version of the VESPUCCI database when available (Moretto et al., submitted paper), and on the other side, the development of new tools, such as those for network reconstruction and functional enrichment analyses. An improvement of the gene annotation, such as the one promoted through the *Vitis* Gene Catalogue within the COST Action INTEGRAPE (integrape.eu), is certainly desirable to make the expansion lists more informative for the biologists.

As a proof of concept of the OneGenE approach, we reconstructed the stilbene synthase gene regulatory network (GRN) and compared it with the available knowledge, represented by both experimental evidence [25,49,50,51,52,53] and in silico analyses [13,49]. The results obtained with OneGenE showed an extended overlap with these previous findings (Table 1), confirming its predictive value. In particular, 50% of the TFs identified by OneGenE as directly connected to *VviSTS* were supported by previous works. The network visualization highlighted a partitioning of the *VviSTSs* around three main hubs, namely *VviMYB14*, the *VviWRKY43*-*53* pair, and the *VviMYB13-15-VviWRKY24* triad (Figure 2A), which may reflect different regulatory cascades. Additional genes have been identified as related to STS, in particular, two C3HC4 type Zinc finger factors (36 and 163), which showed to be highly connected within the *VviWRKY43*-*53* subnetwork, and four JAZ (Jasmonate ZIM domain) TFs located in the *VviMYB14* subnetwork. A closer look at the list of genes related to STSs (Appendix A) revealed that OneGeneE identified many genes involved in JA syntheses, such as allene oxide synthase and 12-oxophytodienoate reductase, and JA signaling, such as two MYC2 TFs and several PR10 genes. The involvement of JA in grapevine response to *Plasmopara viticola*, especially in resistant genotypes such as Regent, which accumulates stilbenoids as a defense strategy [50], has already been pointed out [59], and the accumulation of resveratrol upon treatment of *V. vinifera* cell cultures with JA has been previously reported [60]. Thus, our analysis may contribute to elucidating the signaling cascade connecting JA and stilbenoid synthesis during pathogen infection by proposing novel candidate genes to be investigated. Moreover, some genes related to salicylic acid signaling have also been identified, as expected, by the known cross-talk of JA and SA in the plant defense to this biotrophic pathogen [59].

Once validated, the list of *VviSTS* interacting genes have been inspected to identify genes putatively involved in stilbenoid formation, a class of phytoalexins involved in grapevine defense from pathogens and response to abiotic stress such as UV stress [19]. The compounds with the highest fungitoxic activity include ε-viniferins and trans-pterostilbene, which derive from the oxidative dimerization and methylation of resveratrol, respectively [61]. OneGeneE identified a resveratrol-O-methyltransferase correlated to *VviSTS20*, also negatively correlated to a glycosyltransferase catalyzing the synthesis of piceid, recognized as a storage form of resveratrol without antifungal properties [20]. The enzymes involved in the oxidative oligomerization of resveratrol are not known at the moment. Thus, we attempted to gain insight through the analogy with another class of polyphenols—the monolignols—which have been better characterized in *Arabidopsis*. We noticed that some enzymes contributing to the lignin biosynthetic pathway were also present in the stilbenoid network. These were the LAC and PRX III oxidoreductase enzymatic families involved in monolignol oxidative polymerization [28,31] and the DIR proteins involved in stereospecific positioning of the reactive intermediates [30]. There is evidence that resveratrol units form phenoxyl-radicals upon the action of oxidases such as peroxidase or laccase enzymes, which then condense to form highly polymerized stilbenoids [21,27,62]. Indeed, six *VviPrxIII* and two *LAC* genes were recently identified within QTL regions associated with stilbenoid accumulation [26].

Furthermore, fungal laccases could also drive resveratrol oligomerization or degradation during plant–pathogen interactions [63,64], increasing the complexity of the process. Finally, the involvement of dirigent proteins (DIR), similar to those described for the synthesis of optically active lignans and lignins, has been speculated due to the stereospecificity of stilbene oligomers, but not proven thus far [21,27]. We, therefore, reconstructed the lignin network using the OneGenE expansion lists of six MYB TFs that were proposed as involved in lignin biosynthesis in previous work [33]. Interestingly, we obtained two neatly distinguished lists of isoforms for the three enzymatic families putatively involved in stilbenoid and lignin synthesis (Figure 3 and Figure 4): 16 *VviLAC*, 2 *VviPrx III*, and 4 *VviDIR* genes exclusively associated with the lignin network, and 14 *VviLAC*, 9 *VviPrx* and 5 *VviDIR* genes exclusively associated to the stilbenoid network. It is worth pointing out that *VviPrxIII07c*, identified in the lignin network, is the homolog of *AtPrx64*—based on the phylogenetic reconstruction of the *VviPrxIII* gene family [26]—characterized as involved in sclerenchyma lignification [31]. Conversely, *VviPrxIII23b* and *VviPrxIII23c* identified in the stilbenoid network belong to the class 23, as does *VviPrxIII23a,* a dimeric stilbenoid previously found within a QTL region associated with ω-viniferin [26]. The other seven *VviPrx* genes connected to *VviSTS* and belonging to classes 01 and 22 represent novel candidates for stilbenoid formation to be further functionally characterized.

For the *LAC* and *DIR* families, a genome-wide characterization was carried out using the newly released assembly (12X.v2) and annotation (VCost.v3) of the grapevine genome [42], which allowed to retrieve additional genes and refine them based on RNA-seq data (Table 2 and Table 4). Phylogenetic analysis performed on the manually curated sequences revealed interesting features of these families (Figure 5 and Figure 6), especially for the *LAC* one. They are composed of 76 members, 52 of which are localized on chromosome 18. They are clustered into two groups, named eight and nine, which do not have any *Arabidopsis* homologs (Figure 5, Table 3). We also analyzed the *LAC* family of *Citrus sinensis*, formed by 24 isoforms clustering into seven groups as *Arabidopsis* (not shown), replicating the results from a previously published phylogenetic study [55]. Integrating this analysis with OneGenE results, it appeared that all the stilbenoid-related *LAC* isoforms belonged to group 8, while those identified in the lignin network belonged to groups 1, 2, and 3, and *VviLAC60* to group 8 (Figure 5). Five of the group 8 *LAC* genes, namely *VviLAC38-41* and *VviLAC46*, are located within the *Rpv3.3* locus (as defined by the VVIN16 and UDV737 markers [65]) and were all directly connected to *VviSTSs*. This result supports the observed relationship between this major locus for *P. viticola* resistance present in European resistant grapevines [65] and the accumulation of stilbenoids upon downy mildew infection [26,50,66]. The functional divergence of group 8 and 9 *LAC* genes was coherent with the observation that Arabidospsis and Citrus, which did not synthesize stilbenes, did not undergo this family expansion in their genome. Several pseudogenes named *VviMCO-Like* after our analysis are located in the same region of chromosome 18, suggesting several events of tandem duplication. The OneGenE analysis did not find *VviLACs* of group 9 as associated with *VviSTS*. However, it is interesting to report that group 9 *VviLAC64* and *VviLAC66* are located within the QTL region (centered on the VMC8B5 marker) associated with *ε-*viniferin, a dimeric stilbenoid, and therefore putatively involved in its oligomerization (Appendix A [26]. *DIR* isoforms clustered with *Arabidopsis* ones, despite a limited expansion of the bd2 group (Figure 6). Phylogenetic groups reflected close localization on different chromosomes, suggesting that expansion was gained by gene tandem duplication. Only one isoform appeared distinct from all the others and is located alone on chromosome 4, *VviDIR41*. This orphan gene is associated with stilbenoids, as bd2 isoforms, while those associated with lignin synthesis are in groups a, and bd1, and *VviDIR8* in group bd2.

## 5. Conclusions

We have presented the application of a causality-based approach to reconstruct association gene networks in the grapevine. OneGenE precomputed a ranked list of directly related genes for each gene of the *V. vinifera* genome, which are accessible for download from a dedicated website. Two tools have also been made available for reconstructing gene networks based on a selection of genes of interest for the user. The stilbenoid network, reconstructed from *VviSTS* genes, has been thoroughly investigated both in its regulation, finding known and novel TFs, and metabolic pathway, identifying genes potentially involved in resveratrol modification and oligomerization. The *LAC* and *DIR* family have been characterized on a genome-wide scale and revealed a functional specialization in polyphenols oxidative oligomerization belonging to different pathways, such as those of stilbenoids and lignin biosynthesis. OneGenE proved to be an efficient method able to cope with the complexity of transcriptomic data to identify direct connections among genes and to propose novel candidates for experimental investigations.

## Figures and Tables

**Figure 1 biomolecules-11-01744-f001:**
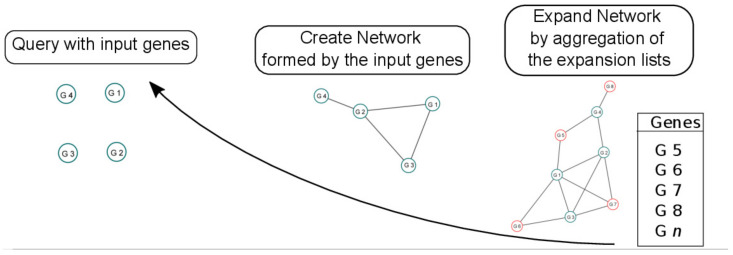
A general pipeline of the analysis to reconstruct a gene network.

**Figure 2 biomolecules-11-01744-f002:**
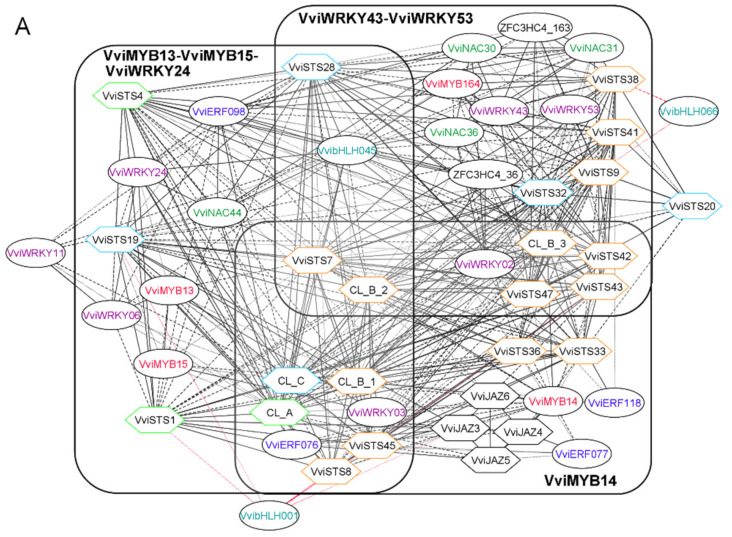
The STS GRN. (**A**) Partitioning of the STS GRN into three overlapping subnetworks, enclosed in rectangles, centered on *VviMYB14*, *VviWRKY43*-*53,* and *VviMYB13*-*15-VviWRKY24.* The STS hexagonal nodes border is colored in green, orange, and light blue according to the phylogenetic groups A, B, and C as defined in [22]. The Edge line type (solid, dashed, or dotted) represents edge weight classes based on the mean relative frequency (decreasing from 1 to 0.5). Red edges correspond to negative values of Pearson’s correlation coefficient, thus joining anticorrelated genes. (**B**) The JAZ TFs subnetwork was extracted from the STS GRN. The JAZ TFs adjacent nodes have been extracted from the STS GRN to better visualize this part of the network.

**Figure 3 biomolecules-11-01744-f003:**
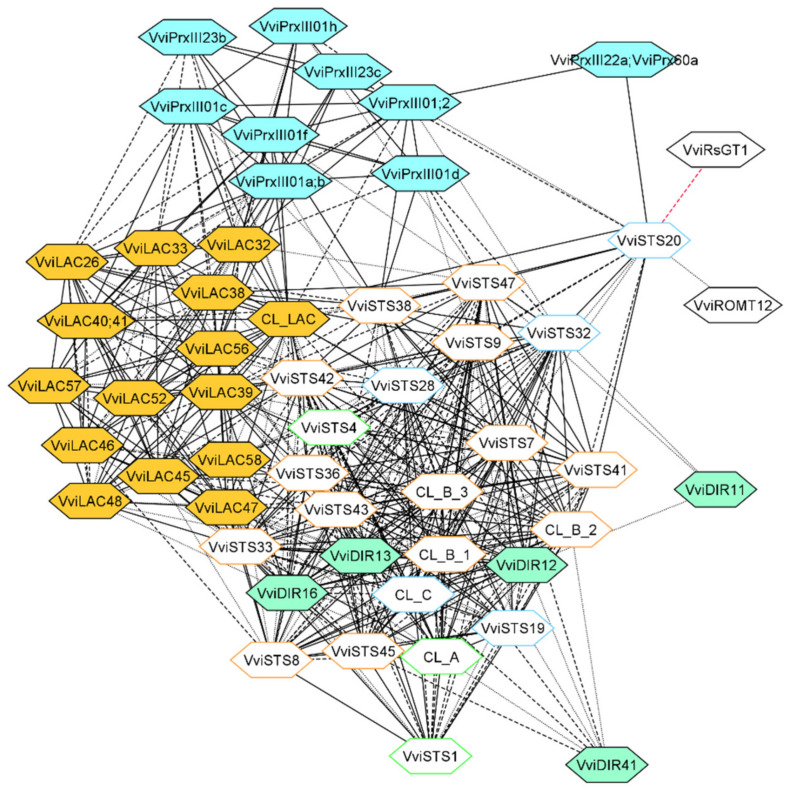
The LAC, PRX, and DIR isoforms of the stilbenoid network. LAC, DIR, and class III PRX nodes are colored in orange, light green, and light blue, respectively. The Edge line type and STS nodes border is defined as in Figure 2.

**Figure 4 biomolecules-11-01744-f004:**
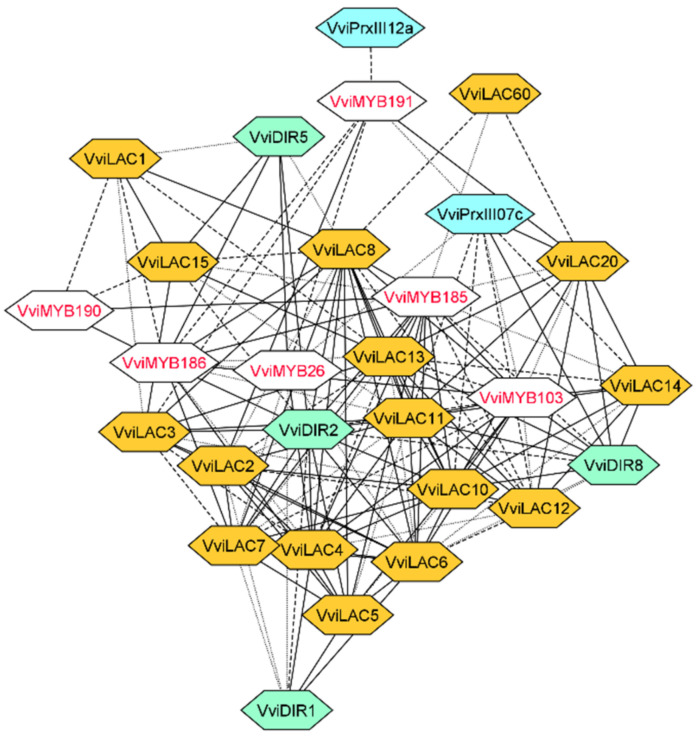
The LAC, PRX, and DIR isoforms of the lignin network. LAC, DIR, and class III PRX nodes are colored in orange, light green, and light blue, respectively. The edge line type and STS nodes border is defined as in Figure 2 and Figure 3.

**Figure 5 biomolecules-11-01744-f005:**
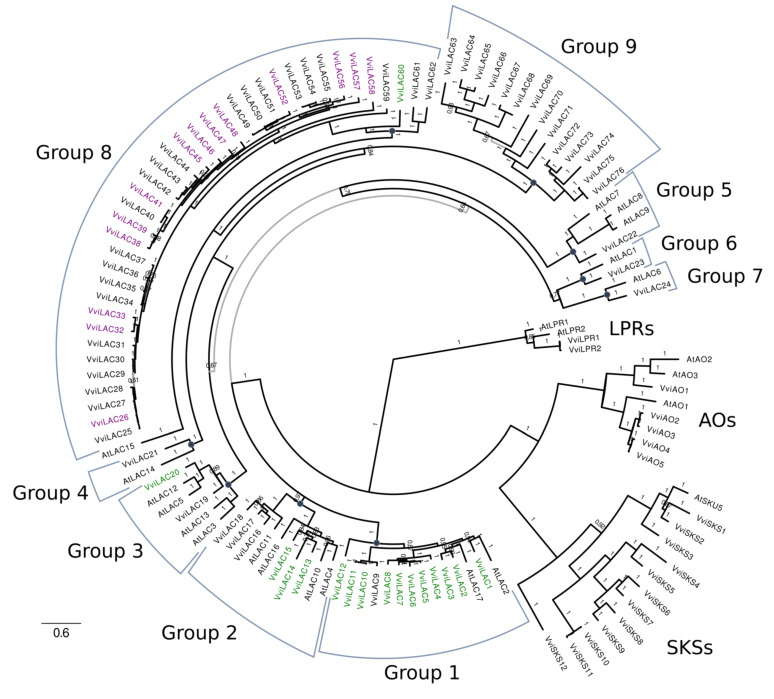
Phylogenetic analysis of the grapevine LAC family and integration with OneGenE results. Bayesian phylogenetic analysis using MrBayes after 3 million generations and average standard deviation of split frequencies of 0.010. LAC genes in the lignin and stilbene OneGeneE networks have green and purple labels, respectively. LAC groups 1–7 have been previously identified in *Arabidopsis* (all contain *Vitis* and *Arabidopsis* sequences). LAC groups 8 and 9 have been newly identified and contain *Vitis* sequences exclusively. This analysis correctly reassigned some putative LACs to the ascorbate oxidases (AO) and SKU5 similar (SKS) families, while two *Vitis* genes homologous to the two low phosphate root (LPR) genes in *Arabidopsis* were used as an outgroup.

**Figure 6 biomolecules-11-01744-f006:**
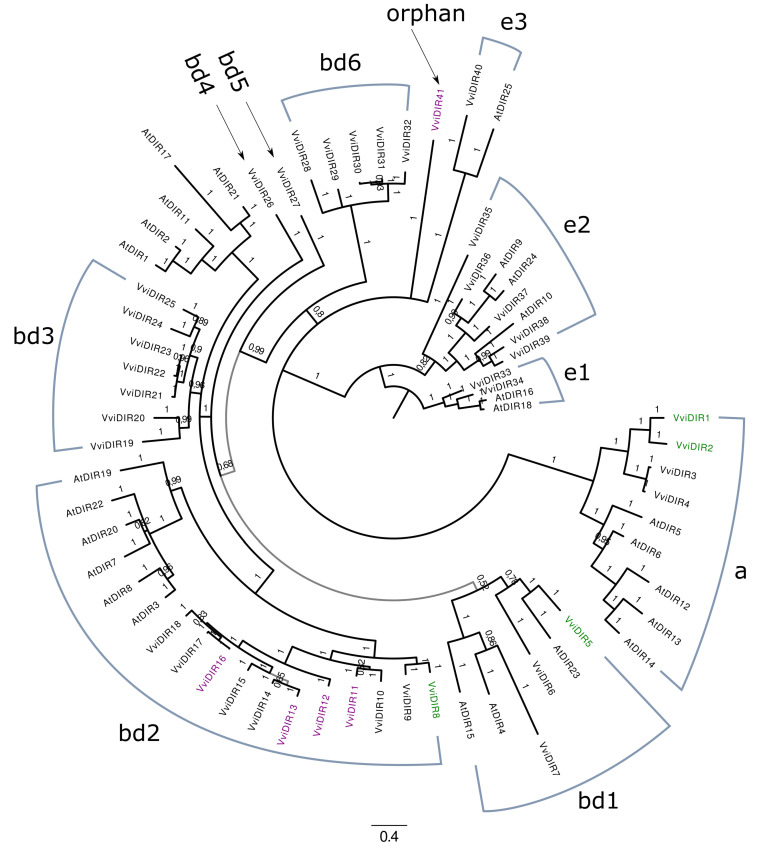
Phylogenetic analysis of the grapevine DIR family and integration with OneGenE results. Bayesian phylogenetic analysis using MrBayes after 1 million generations and average standard deviation of split frequencies of 0.0075. Subgroup e1 was used to root the tree. *DIR* genes in the lignin and stilbene OneGeneE networks have green and purple labels, respectively.

**Table 1 biomolecules-11-01744-t001:** List of the transcription factors of the STS GRN identified with OneGenE and mentioned in previous works.

N. of Connected *VviSTS*	12X.v1	Gene Name	Reference	N. of Connected *VviSTS*	12X.v1	Gene Name	Reference
15	Vitvi11g00165	VvibHLH045	[49,50]	2	Vitvi17g00025	VviERF118	[49]
14	Vitvi07g00598	VviMYB14	[13,49,51]	2	Vitvi15g00948	VvibHLH066, MYC7E	-
13	Vitvi07g02070	VviERF098	[49,50]	2	Vitvi15g00938	VviMYBPA1	-
9	Vitvi14g01523	VviWRKY43	[13,49,52]	2	Vitvi08g01412	VviRAV5	-
8	Vitvi08g01931	VviHsfB3a	[49,50]	2	Vitvi08g01224	ZFDHHC_8	[52]
7	Vitvi05g01732	VviMYB13	[49]	1	Vitvi14g01617	C2H2FAM_25	-
7	Vitvi17g00309	VviMYB164	-	1	Vitvi11g00715	C2H2FAM_28	-
7	Vitvi06g00007	VviNAC44	[49,52]	1	Vitvi11g00436	CCCHFAM_12	-
7	Vitvi08g00793	VviWRKY24	[49,50,53]	1	Vitvi15g00636	GATA2	-
7	Vitvi13g00620	ZFC3HC4_163	-	1	Vitvi13g01879	GCN5_18	-
6	Vitvi15g01203	ERF2_1	[50]	1		IAA22D_2	-
6	Vitvi04g02175	VviJAZ3	-	1	Vitvi06g00167	JMJD5	-
6	Vitvi19g02272	VviNAC31	[49]	1	Vitvi15g00736	VviLBDIa2 LBD19	-
6	Vitvi12g00076	VviNAC36	[49,50]	1	Vitvi13g00085	VviLBDIc3 LBD21	[50,52]
5	Vitvi05g00523	GCN5_7	[50]	1	Vitvi00g01060	VviLBDIc5	-
5	Vitvi08g01525	HMGIY_1	[52]	1	Vitvi01g01921	MYB3R1_5	-
5	Vitvi01g01287	VvibHLH001, BIM3	-	1	Vitvi14g00217	NFYB5_3	-
5	Vitvi05g01733	VviMYB15	[13,49,50,51]	1	Vitvi18g01322	RGL2_1	-
5	Vitvi04g00133	VviWRKY06	[50]	1	Vitvi01g00185	SAP7_3	-
5	Vitvi17g00556	VviWRKY53	[49,50,51]	1	Vitvi13g00311	SCL14_1	[49,52]
4	Vitvi10g00053	JMJD1B_5	[50]	1	Vitvi07g01762	TFB3_4	-
4	Vitvi19g02273	VviNAC30	[52]	1	Vitvi07g01529	VvibHLH038	-
4	Vitvi17g00801	ZFC3HC4_103	-	1	Vitvi15g01027	VvibZIP40	-
3	Vitvi07g02168	MTERF_10	-	1	Vitvi05g01722	VviERF109	-
3	Vitvi01g00940	VviWRKY02	[13,49,50]	1	Vitvi15g01206	VviERF073	[49]
3	Vitvi01g01680	VviWRKY03	[49,50,52]	1	Vitvi19g01669	VviMYB139	[49,52]
3	Vitvi04g00760	VviWRKY11	[49]	1	Vitvi07g03055	VviMYB148	[49]
3	Vitvi19g00155	ZFC3HC4_158	[52]	1	Vitvi19g01564	VviNAC29	[52]
3	Vitvi12g00220	ZFC3HC4_36	[50]	1	Vitvi04g00756	VviWRKY10	-
2	Vitvi10g00572	-	-	1	Vitvi06g01574	VviWRKY15	[50]
2	Vitvi13g01958	ERUNK_7	-	1	Vitvi10g00063	VviWRKY29	[49,53]
2	Vitvi09g00064	VviJAZ4	-	1	Vitvi13g00189	VviWRKY40	[52]
2	Vitvi10g00826	VviJAZ5	-	1	Vitvi16g01213	VviWRKY51	[50]
2	Vitvi10g01879	VviJAZ6	-	1	Vitvi05g00368	ZFC3HC4_164	-
2	Vitvi16g00349	VviERF077	-	1	Vitvi08g00923	ZFC3HC4_43	-
2	Vitvi02g01780	VviERF076	[50]	1	Vitvi16g02032	ZFC3HC4_45	-

**Table 2 biomolecules-11-01744-t002:** Gene model curation of putative V. vinifera *LAC* genes.

Source	Initial Genes	Corrected CDS	Split Genes	Merged Genes	Curated Gene Models	Identified as Pseudogenes or Non-Laccases	Final Gene Models
Present in 12X.v1	77	16	2	3	76	14	62
HMMER search in 12X.v2	44	4	0	0	44	11	33
Total	121	20	2	3	120	25	95

**Table 3 biomolecules-11-01744-t003:** Comparison between *Vitis* and *Arabidopsis* groups and subgroups cardinality.

Species	LAC (Group 1 to 7)	LAC (Group 8 and 9)	Total LACs	Low Phosphate Root	Ascorbate Oxidases	SKU5 Similar
*Vitis vinifera*	24	52	76	2	5	12
*Arabidopsis thaliana*	17	0	17	2	3	19

**Table 4 biomolecules-11-01744-t004:** Gene model curation of putative *V. vinifera* DIR genes.

Source	Initial Genes	Corrected CDS	Split Genes	Merged Genes	Curated Gene Models
Present in 12X.v1	32	6	0	0	32
HMMER search in 12X.v2	9	2	0	0	9
Total	41	8	0	0	41

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
