# Peer review of "Vitis OneGenE: A Causality-Based Approach to Generate Gene Networks in Vitis vinifera Sheds Light on the Laccase and Dirigent Gene Families"

_biomolecules, 2021, doi:10.3390/biom11121744_

Round 1

Reviewer 1 Report

This manuscript described the application of OneGenE in data mining that finds correlated genes for each gene of grapevine genome. As presented by the authors, the tool can accelerate knowledge discovery by suggesting new candidates for functional characterization and application in breeding programs by network analysis. Some suggestions as follow:

  1. I have tried to log in the website and run the analysis of STS network acording to the tutorial, 'Create network' can be done well. But 'expand network' did not get the results, maybe it is affected by the speed of websit, maybe also because of the amount of computation.
  2. Figure 3 and 4,  please indicate the meaning of the color symbol.The table should be changed to a three-line grid format.
  3. Other changes can be found in the revised manuscript.
  •  

Reviewer 2 Report

Authors need to check and edit some typing errors.

Authors should also include more plant species (from base plants to flowing plants) to check whether the model is correct following the plant evolution of the two selected pathways.

Reviewer 3 Report

In the manuscript entitled "Vitis OneGenE, a causality-based approach to generate gene networks in Vitis vinifera, sheds light on the laccase and dirigent gene families" authors describe the application of OneGenE causality based data mining tool that finds directly correlated genes for each gene of a genome. The results obtained were compared with other published experimental data. The authors used the secondary metabolic pathways of stilbens and lignin as an example to confirm the workability of their analysis tool. The authors have done a lot of bioinformatic work that deserves attention. Each stage of the creation of OneGenE approach is described in detail. I think that this research tool will be useful for the study of gene networks.

I consider, that this work is worthy of publication in the journal Biomolecules. I recommend accepting this manuscript for publication after a minor edit.

  • 2. Mark Figure 2A and 2B.
  • It would be great to enlarge the drawings or inscriptions on them. Very small designations because of this information is poorly perceived.
  • Why in this paper the authors limited themselves to the study of the network of stilbens and lignin? A lot has been studied in this area and it is difficult to obtain data that would be fundamentally new. I understand that you are proposing a new tool for analyzing gene networks, but I would like to see more novelty in the study of the biosynthesis of stilbens or look at the gene network of the entire phenylpropanoid pathway of grapes.
  • As I understand it, OneGenE data is a tool for analyzing Vitis vinifera. Is it possible to interpret the results of Vitis vinifera on other species of grapes?
  • You have identified genes that could potentially be involved in the modification and oligomerization of resveratrol. There are other works that make their assumptions (for example, doi:10.3389/fpsyg.2019.00234). However, at the moment these are just assumptions. Is it possible to convincingly prove the role of these genes in the oligomerization of stilbens? Of course, most likely many enzymes are involved in this process, but it would be interesting to identify at least a few that are involved in the conversion of resveratrol to viniferin.

Round 2

Reviewer 1 Report

The revised manuscript have been well improved. I hoped that the program can be further optimized and make it easy to be used.